# Spatio-temporal patterns of the crustacean demersal fishery discard from the south Humboldt Current System, based on scientific observer program (2014–2019)

**Mauricio F. Landaeta**[1,2,3]*, **Carola Hernández-Santoro**[4,5], **Francesca V. Search**[2], **Manuel I. Castillo**[3,6], **Claudio Bernal**[4], **Sergio A. Navarrete**[2,7], **Evie A. Wieters**[2,7], **Ricardo Beldade**[2,7], **Ana Navarro Campoi**[2,7], **Alejandro Pérez-Matus**[2,8]

1 Laboratorio de Ictiología e Interacciones Biofísicas (LABITI), Instituto de Biología, Facultad de Ciencias, Universidad de Valparaíso, Valparaíso, Chile, 2 Millennium Nucleus for the Ecology and Conservation of Temperate Mesophotic Reef Ecosystem (NUTME), Las Cruces, Chile, 3 Centro de Observación Marino para Estudios del Ambiente Costero (COSTA-R), Universidad de Valparaíso, Viña del Mar, Chile, 4 Instituto de Fomento Pesquero (IFOP), Valparaíso, Chile, 5 Programa Doctorado en Ciencias del Mar y Biología Aplicada, University of Alicante, San Vicente del Raspeig, Spain, 6 Laboratorio de Oceanografía Física y Satelital (LOFISAT), Escuela de Biología Marina, Universidad de Valparaíso, Viña del Mar, Chile, 7 Estación Costera de Investigaciones Marinas, Pontificia Universidad Católica de Chile, Las Cruces, Chile, 8 Departamento de Ecología, Subtidal Ecology Laboratory, Estación Costera de Investigaciones Marinas, Facultad de Ciencias Biológicas, Pontificia Universidad Católica de Chile, Santiago, Chile

* mauricio.landaeta@uv.cl

**Data Availability Statement:** The data underlying the results presented in the study are available

## Abstract

This study summarises six years of spatio-temporal patterns of the discarded demersal community fauna recorded by onboard scientific observer program for both artisanal and industrial crustacean fisheries between 2014 and 2019, from mesophotic to aphotic depths (96 to 650 m) along the southern Humboldt Current System (28–38˚S). In this period, one cold and two warm climatic events were observed during the austral summer 2014, 2015–2016 (ENSO Godzilla), and 2016–2017 (coastal ENSO), respectively. Satellite information showed that Chlorophyll-a concentration varied seasonally and latitudinally, associated with upwelling centres, while equatorial wind stress decreased southward of 36˚S. Discards were composed of 108 species, dominated by finfish and molluscs. The Chilean hake *Merluccius gayi* was dominant and ubiquitous (occurrence, 95% of 9104 hauls), being the most vulnerable species of the bycatch. Three assemblages were identified: assemblage 1 (~200 m deep), dominated by flounders *Hippoglossina macrops* and lemon crabs *Platymera gaudichaudii*, assemblage 2 (~260 m deep), dominated by squat lobsters *Pleuroncodes monodon* and *Cervimunida johni* and assemblage 3 (~320 m depth), dominated by grenadiers *Coelorinchus aconcagua* and cardinalfish *Epigonus crassicaudus*. These assemblages were segregated by depth, and varied by year, and geographic zone. The latter represented changes in the width of the continental shelf, increasing southward of 36˚S. Alpha-diversity indexes (richness, Shannon, Simpson, and Pielou) also varied with depth and latitude, with higher diversity in deeper continental waters (>300 m), between 2018–2019. Finally, at a spatial scale of tens of kilometres, and a monthly basis, interannual variations of biodiversity occurred in the demersal community. Surface sea temperature, chlorophyll-a, or wind stress

from Instituto de Fomento Pesquero (https://www.ifop.cl/busqueda-de-informes/). Documents and data are available using División de Investigación: Pesquerias, and Tipo de Proyecto: Pesquerias.

**Funding:** This work was partially funded by the Millenium Nucleus for Ecology and Conservation of Temperate Mesophotic Reef Ecosystems (NUTME) grant NCN19_056 to APM, EAW, SAN, RB, PSA and MFL The funders had no role in study design, data collection and analysis, decision to publish, or preparation of the manuscript."

**Competing interests:** The authors have declared that no competing interests exist.

did not correlate with discarded demersal fauna diversity of the crustacean fishery operating along central Chile.

## Introduction

The coastal ocean of the productive Humboldt Current System (HCS) supports many economically and socially important fisheries operating at the shelf break, over the continental shelf, and in nearshore habitats along Chile and Peru. Across the continental shelf and shelf break, large demersal fisheries, such as the red squat lobster (*Pleuroncodes monodon*), yellow squat lobster (*Cervimunida johni*), deep-water shrimps (*Heterocarpus reedi* and *Haliporoides diomedeae*), and Chilean hake (*Merluccius gayi*), have been operating with bottom trawling since the 1950s [1]. It is common in all trawling fisheries to find high levels of bycatch (portion of the catch that is not comprised of fishery's target species) or discarded species (portion of the catch that is thrown back into the sea), considered a waste of natural resources and a contribution to the depletion of the stocks that are under high fishing pressure [2]. Indeed, although techniques have been developed to reduce bycatch and discards [3, 4], the inevitable inclusion of non-target or 'undesired' species in the trawls is considered one of the most important impacts of fisheries in marine ecosystems [5]. Here we provide a baseline description of the discarded community associated with demersal crustacean Chilean fisheries.

Like many other large-scale fisheries worldwide [6, 7], there is now widespread evidence that some, if not most, of these industrial fisheries are over-exploited or outright collapsed [8], despite efforts to manage them sustainably. Trophic models estimate that fishing removes about 15% of the primary production from central Chile, extracting predominantly from intermediate to low trophic levels of the food web [9]. The reasons for unsustainable practices, inappropriate management strategies, and inconclusive assessments of the true state of wild populations are multiple and still hotly debated [10, 11]. Two core issues are 1) the predominant 'monospecific approach' to fisheries, which has curtailed our capacity to visualize fished species within complex interactive communities to advance toward an ecosystem-level approach [12], and 2) our still primordial understanding about how these systems respond to natural and anthropogenic environmental variation [13]. Indeed, the lack of scientific information about exploited ecosystems and their spatial and temporal variability over scales relevant to fisheries management has hampered efforts to implement better management and adaptation strategies in the face of climate change [14].

For demersal fauna, it is well known that regional and basin-scale meteorological and oceanographic processes, as well as their interactions with topography, can have an impact on population [15] and community levels [16]. The demersal community of the HCS has been described using prospective surveys, which is dominated by the Chilean hake *Merluccius gayi*, bigeye flounder *Hippoglossina macrops*, and nylon shrimp *Heterocarpus reedi* [14, 17, 18]. Three to four groups of species assemblages have been previously identified for demersal fishes and crustaceans [14, 17, 19] based on the depth distribution of the species and trophic interactions. However, intra-annual variability and its dependence on surface physical processes are poorly understood.

The Chilean demersal fauna interacts with the Equatorial Subsurface Water (ESSW) and the Antarctic Intermediate Water [17], and its richness and variation have been related to the high concentration of nutrients and low dissolved oxygen in these water masses [20]. The Peru-Chile poleward undercurrent that flows at depths between 100 and 400 m (core at ~220

m depth), transports ESSW southward, while the Chile-Peru Coastal current, the coastal branch of the Humboldt current, moves the Sub-Antarctic Surface Waters (SASW) northward [21]. During upwelling season (September–March, with a peak in December–January), strong southerly winds induce upwelling of the ESSW at the coast and offshore displacement of the surface layer by Ekman transport, pushing equatorward and offshore the SASW [22]. Conversely, during autumn-winter, the SASW is generally closer to shore and the ESSW tends to deepen toward the continental slope [23]. The seasonal cross-shore displacement of these water masses has been connected to the spatial distribution of the demersal community [24]. During austral spring, when the equatorward winds intensify and upwelling events increase [25], demersal organisms, such as *M. gayi*, *H. macrops*, and the squat lobster *Pleuroncodes monodon*, spawn at depths below 50 m [26–28]. Their larvae are transported onshore via the subsurface flow that compensates the surface offshore Ekman layer during upwelling events [26–28]. At interannual and decadal scales, El Niño-Southern Oscillation (ENSO) and the Pacific Decadal Oscillation (PDO), may negatively impact population growth of the demersal fauna, such as the cardinalfish *Epigonus crassicaudus* living at depths of 300–550 m [29].

Our knowledge of the discard composition and structure, and its relationship with environmental factors is still poor for most fisheries in Chile, and especially for the demersal crustacean fishery. Here we take advantage of the fisheries scientific observer program implemented by the Chilean *Instituto de Fomento Pesquero* (IFOP) to provide baseline information about the composition, as well as patterns of spatial and temporal variation, of the accompanied fauna captured and discarded by the artisanal and industrial crustacean fisheries. Our goals were to characterize patterns of assemblage structure and diversity, describe their latitudinal and bathymetric distribution across central Chile, and evaluate their association with environmental variability at seasonal- and interannual scales. We expect that this information can be used as a baseline to assess future changes and inform managers about the fishery impacts on bycatch species.

## Materials and methods

### Biological data

A total of 9104 commercial hauls from 1563 sampled trips carried out between 28.17˚S and 37.92˚S (Fig 1 and Table 1), from January 2014 to December 2019, onboard the artisanal and industrial demersal crustaceans fleet operating off central Chile were studied. The Chilean crustacean fisheries are open throughout the year, except for September, when a reproductive ban occurs, since 2015. The sampled trips in which scientific observers collected data were randomly selected and represented between 13.5 to 25.1% of the total trips by the crustacean demersal fisheries. Differences between industrial and artisanal commercial bottom trawling fleets are minimal (mean boat length, 16.8 and 21.1 m; mean engine power, 450 and 525 HP, respectively), utilizing the same dimensions of the fishing gear, a net with two panels of 28.8 m headrope and 32 m footrope with sweeps and bridles of 1 and 5 m, respectively. The nets were made of knotted polyethylene; 80 mm mesh size on the upper panel (72 mm inner mesh size), 54 mm in the lower panel (47 mm inner mesh size) and 56 mm at the cod-end (47 mm inner mesh size) [30] and therefore, the type of fleet was not considered in further analyses. Hauls were made during day and night at depths between 96 and 650 m and lasted between 0.16 and 10.5 hours (mean ± SD, 2.38 ± 0.84 hours).

The catch per haul of the discarded species were quantified and identified to the lowest possible taxonomic level by experienced onboard scientific observers. A detailed explanation of the process in available at https://www.ifop.cl/nuestro-que-hacer/la-investigacion-pesquera/depto-de-evaluacion-de-pesquerias/proyectos-de-descarte/, leaded by the Instituto de

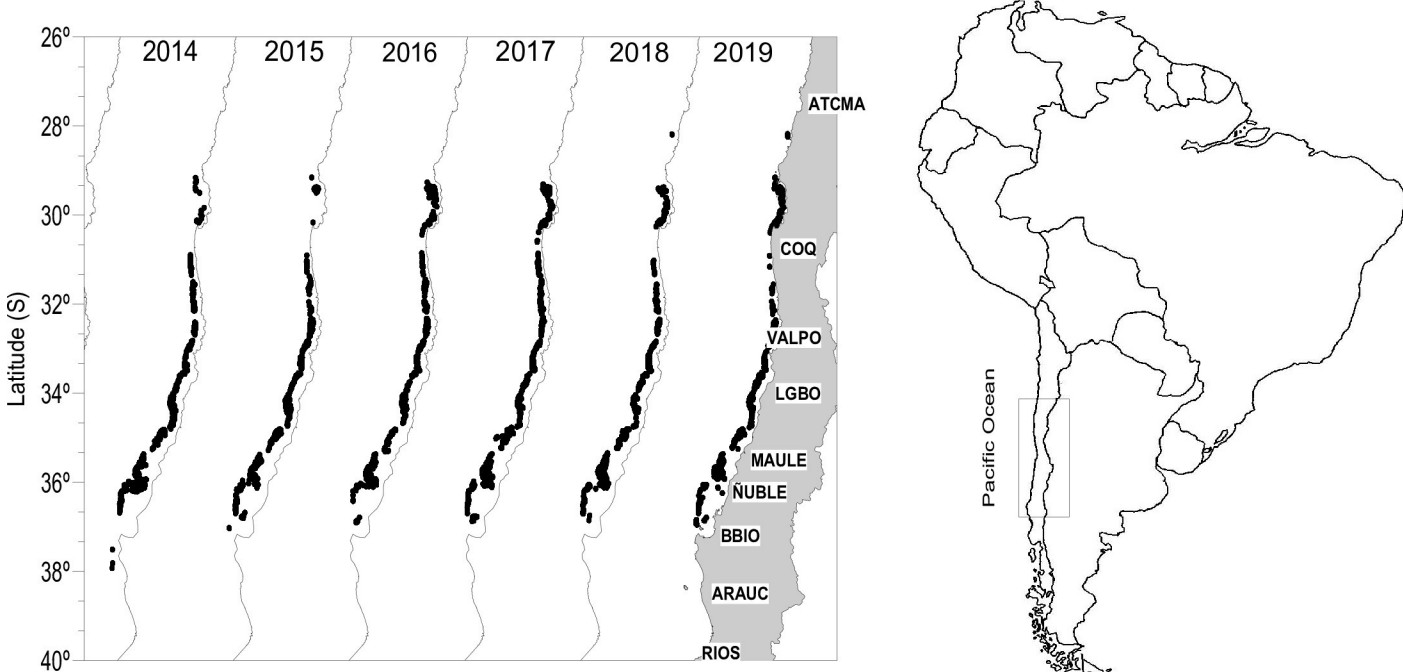

**Fig 1. Spatio-temporal variation of the trawls (black dots) of the artisanal and industrial demersal crustacean fleet, obtained by the scientific onboard observers between 2014–2019.**

Fomento Pesquero (IFOP, Chile). The total weight of each species was obtained as the product of the number and the average weight of standard trays (55 cm x 40 cm x 25 cm) [30]. Then, data was standardized to biomass density (kg km$^{-2}$) for each discarded species and expressed as catch per unit area (CPUA) for each haul.

The cumulative species curves were constructed utilizing the function 'specaccum' from the package 'vegan' for R [31], using the exact estimation method. To determine the co-occurrence of the main discarded species, species representing over 2% of the frequency of occurrence were used, thus avoiding rare species in the analysis [14]. For exploring the spatio-temporal variation in catch composition, CPUA was transformed with log(x+1) to reduce the relative importance of dominant species (e.g., Chilean hake), and non-metric multidimensional scaling (nMDS) was used for ordination analysis, using the metaMDS function of the 'vegan' package [31], based on the Bray-Curtis similarity index. Then, and due to the ubiquitous distribution and abundance of the Chilean hake, a new nMDS was done extracting this species from the analysis. These analyses consider depth strata (90–200, 200–300, 300–400, 400–500, >500 m) and year (2014–2019) as factors. The components of the nMDS were classified using a beta-flexible clustering method with a recommended β = -0.25, using the 'vegan' package [31, 32], since it generally produces recovery rates that are competitive with the group average when outliers are not present [32, 33].

The indicator value (IndVal) composite index [34] was used to detect the species that best characterized different assemblages. IndVal considers the abundance and relative frequency of the different taxa in each sample to assign an indicator value and a probability derived from a Monte Carlo permutation (n = 9999), using the 'vegan' package [31].

The depth distribution of each group detected by the clustering and nMDS methods, was compared with a two-way nested ANOVA, considering latitude (from 28˚S to 36˚S, and 36˚S southward, according to the width of the continental shelf) and year (2014–2019) (nested into

**Table 1. Composition of the main species representing the discarded fauna associated with the artisanal and industrial fleet of the crustacean demersal fishery operating off central Chile (28° to 38°S) from 2014 to 2019.**

| Taxonomic group | Scientific name | Abbreviation | Occurrence (%) | Dominance (% of total CPUA) |
|---|---|---|---|---|
| Teleostei | *Merluccius gayi* | MerGay | 95.0 | 42.77 |
| Teleostei | *Hippoglossina macrops* | HoMac | 60.2 | 6.73 |
| Malacostraca | *Platymera gaudichaudii* | PyGau | 57.2 | 3.92 |
| Malacostraca | *Cancer porteri* | CPort | 54.5 | 3.32 |
| Teleostei | *Coelorinchus aconcagua* | CAcag | 34.0 | 7.23 |
| Malacostraca | *Pleuroncodes monodon* | PMono | 19.7 | 15.32 |
| Malacostraca | *Cervimunida johni* | CJohn | 18.6 | 10.78 |
| Teleostei | *Epigonus crassicaudus* | Eras | 17.5 | 0.38 |
| Mollusca | *Muusoctopus eicomar* | MoEic | 16.0 | 0.02 |
| Elasmobranchii | *Aculeola nigra* | ANigr | 13.2 | 0.83 |
| Teleostei | *Coelorinchus chilensis* | CoChil | 13.0 | 2.17 |
| Elasmobranchii | *Psammobatis scobina* | PScob | 12.6 | 0.42 |
| Elasmobranchii | *Centroscyllium granulatum* | CeGran | 10.5 | 0.49 |
| Malacostraca | *Heterocarpus reedi* | HReed | 9.1 | 3.32 |
| Malacostraca | *Libidoclaea granaria* | LiGran | 8.5 | 0.14 |
| Teleostei | *Genypterus maculatus* | GMacu | 7.4 | 0.19 |
| Elasmobranchii | *Bythaelurus canescens* | BaCan | 7.3 | 0.25 |
| Elasmobranchii | *Zearaja chilensis* | ZChil | 6.5 | 0.25 |
| Agnatha | *Eptatetrus polytrema* | EPoly | 5.0 | 0.06 |
| Teleostei | *Prolatilus jugularis* | PaJug | 4.0 | 0.06 |
| Teleostei | *Guttigadus kongi* | GKong | 3.9 | 0.03 |
| Malacostraca | *Haliporoides diomedeae* | HDiom | 3.3 | 0.06 |
| Malacostraca | *Pterygosquilla armata* | PArm | 3.1 | 0.01 |
| Mollusca | *Dosidicus gigas* | DGiga | 3.0 | 0.14 |
| Elasmobranchii | *Psammobatis rudis* | PRud | 2.9 | 0.43 |
| Mollusca | *Muusoctopus longibrachus* | MLong | 2.8 | 0.02 |
| Elasmobranchii | *Centroscyllium nigrum* | CeNig | 2.5 | 0.10 |
| Malacostraca | *Lophorochinia parabranchia* | LoPara | 2.2 | 0.09 |
| Elasmobranchii | *Bathyraja peruana* | BPeru | 2.0 | 0.03 |
| Anthozoa | *Hormathia pectinata* | Horte | 2.0 | 0.01 |

assemblages) as factors. Additionally, to determine the differences in species composition, a permutational multivariate analysis of variance (PERMANOVA, 9999 permutations) was performed, using the "adonis" routine from the "vegan" package [31]. The corrected Akaike Information Criterion (AICc) was used for the selection of the factors with the best fit. We examined the similarity of species composition among years, latitude, season, and depth of the set.

Alpha-diversity indices, such as the species richness ($S$), Shannon-Wiener ($H'$), Simpson ($1-\lambda$), and Pielou's evenness index ($J'$) were calculated based on the CPUA of all taxa identified (without exclusions), using the "vegan" package [31]. They were used to investigate the spatio-temporal trend of the biodiversity of the discarded community, considering latitude vs depth by year, and latitude vs time (months). The patterns were interpolated using kriging in Hovmöller diagrams with Surfer 10.

## Environmental conditions

The absence of spatially and temporally extensive data for subsurface conditions restricted our environmental characterization to surface analyses. Three satellite data sets (sea surface

temperature, chlorophyll-a, and surface winds) were used to characterize the environmental variability along the northern-central Chilean coastline. The study area stretched from 28˚S to 40˚S, covering the same latitudinal gradient as the biological data, as well as the important topographic and coastline features around the Arauco Gulf (~37˚S). To study coastal processes occurring over the continental shelf and shelf break, all data sets were limited to a band, measuring ~100 nautical miles (nm) from the coastline. To determine long-term temporal trends in the area, as well as the potential influence of regional and basin-scale physical processes on the demersal discarded fauna, a period between 2003 to 2020 was analyzed. Because most of the fauna, particularly finfish and elasmobranchs, are long-lived species, oceanographic data and putative processes occurring before the biological time series was considered, may have an impact on life history, abundance, and distribution of these species.

Sea surface temperature (SST) regional variability was obtained using satellite data derived from the Multi-scale Ultra high Resolution (MUR) data set (https://registry.opendata.aws/mur/), 1 x 1 km gridded data was downloaded for the period between 2003 and 2020. To explore the time series variability, daily SST images from the global gridded data were cropped to a coastal region (first 100 nm). This subsample was then used to estimate the SST anomalies (SSTA), which was obtained by calculating the annual and semi-annual signal contribution using a harmonic analysis [35] and subsequently filtered. SSTA variability (without the annual cycle) along the region was described by an Empirical Orthogonal Function (EOF) analysis, providing a spatio-temporal study of the dominant modes.

Monthly averaged chlorophyll-a (Chl-a) satellite $4 \times 4$ km gridded data sets, from 2003 to 2020, were obtained from the Moderate-Resolution Imaging Spectroradiometer dataset (MODIS project) and downloaded from the OceanColor website [36]. Hourly surface winds, with a spatial resolution of 0.25˚ x 0.25˚, were obtained from the ERA5 climate data set (https://cds.climate.copernicus.eu/). For use in this study, wind data was subsequently averaged daily, and meridional wind-stress was calculated using the dimensionless drag coefficient was calculated using the formula proposed by Yelland and Taylor [37].

## Modelling

Finally, Generalized Additive Models (GAMs) were used to model the effects of environmental predictors (Depth, Longitude, Latitude, Year, SST, SSTa, Chl-a, and wind-stress) on the biodiversity of discards, using the packages "MASS" and "Rcmdr" for R. Alpha-diversity indices were calculated monthly for quadrants of one-degree latitude and 27 km offshore, to coincide with the pixel of the ERA5 surface wind data (27 x 27 km). A Poisson distribution was used to model the richness, while a gaussian distribution was used to model the other diversity indices. The fitting of each model was evaluated using the Akaike information criterion (AIC) [38]. Because of the high collinearity (correlation $> 0.7$) between latitude and longitude, models were evaluated separately and then compared using a $F$ test. After selecting the model that better explained the dependent variable, the importance of each factor was evaluated, using partial pseudo-$R^2$ [39].

## Results

### Composition and structure of the discarded demersal community off south HCS

The total catch discarded in 9104 hauls was composed of 108 taxonomic groups (S1 File), dominated by teleost fish (59.9% of total CPUA), followed by Malacostraca (36.9%), and Elasmobranchii (2.7%). The filtered biological matrix was composed of 30 taxa (Table 2), from

**Table 2. Summary of the results of the indicator value composite index (IndVal). sd represents one standard deviation.**

| Assemblage | Dominant species | Occurrence (%) | Mean Depth (m)/sd | Indicator species | stat | P value |
|---|---|---|---|---|---|---|
| 1 | *Hippoglossina macrops* | 98.0 | 196/56 | *Cervimunida johni* | 0.460 | 0.001 |
| | *Merluccius gayi* | 95.7 | | *Heterocarpus reedi* | 0.345 | 0.001 |
| | *Platymera gaudichaudii* | 91.1 | | | | |
| | *Cancer porteri* | 88.3 | | | | |
| | *Pleuroncodes monodon* | 24.8 | | | | |
| | *Cervimunida johni* | 18.3 | | | | |
| | *Coelorinchus aconcagua* | 15.7 | | | | |
| | *Prolatilus jugularis* | 8.1 | | | | |
| 2 | *Merluccius gayi* | 91.6 | 258/79 | | | |
| | *Pleuroncodes monodon* | 30.0 | | | | |
| | *Cervimunida johni* | 28.3 | | | | |
| | *Muusoctopus eicomar* | 21.4 | | No information | | |
| | *Epigous crassicaudus* | 19.2 | | | | |
| | *Heterocarpus reedi* | 14.5 | | | | |
| | *Genypterus maculatus* | 11.9 | | | | |
| | *Haliporoides diomedae* | 7.2 | | | | |
| 3 | *Merluccius gayi* | 97.3 | 326/58 | *Coelorinchus aconcagua* | 0.915 | 0.001 |
| | *Coelorinchus aconcagua* | 89.3 | | *Coelorinchus chilensis* | 0.586 | 0.001 |
| | *Hippoglossina macrops* | 65.7 | | *Psammobatis scobina* | 0.580 | 0.001 |
| | *Platymera gaudichaudii* | 65.2 | | *Centroscyllium granulatum* | 0.561 | 0.001 |
| | *Cancer porteri* | 60.4 | | | | |
| | *Coelorinchus chilensis* | 37.0 | | | | |
| | *Psammobatis scobina* | 35.6 | | | | |
| | *Aculeola nigra* | 35.0 | | | | |

which the Chilean hake *Merluccius gayi*, was the dominant (44.7% of total CPUA) and most frequent discarded species, reaching 95% of occurrence, followed by bigeye flounder *Hippoglossina macrops* (6.7% and 60.2% of dominance and occurrence, respectively), and lemon crab *Platymera gaudichaudii* (3.2% and 57.2%, respectively). Taxonomic groups with greatest compositional representation were the crustaceans (9 taxa), elasmobranchs (9 taxa), and Teleostei (8 taxa) (Table 1).

The cumulative curves of species found as discards showed some interannual variations (Fig 2). In 2014, there was a higher number of species, showing also larger standard deviation, particularly between 50 and 150 trawls. During 2015 and 2016, there were a decrease in the number of species, as well as its deviation, while between 2017–2019, the number of species decreased slightly, and the standard deviation increased for large number of trawling (between 300 and 800) (Fig 2).

The beta-flexible clustering analysis and the first two axes of the nMDS (stress = 0.119) suggested the presence of three assemblages (Fig 3A). Assemblage 1 (red in Fig 3B) was dominated by the bigeye flounder *H. macrops* and *Platymera gaudichaudii*. While assemblage 2 (green in Fig 3) corresponded mainly to non-target crustaceans (e.g., squat lobsters *P. monodon* and *C. johni* for the fishery of *H. reedi*, *C. johni* and *H. reedi* for the fishery of *P. monodon*, and *P. monodon* and *H. reedi* for the fishery of *C. johni*), as well as the deep-sea octopus *Musooctopus eicomar*. Assemblage 3 (blue in Fig 3B) was dominated by teleosts and elasmobranchs, such as *Coelorinchus aconcagua*, *Epigonus crassicaudus*, *Psammobatis scobina*, and *Aculeola nigra*. The main indicator species according to the IndVal analysis were *C. johni* and *Heterocarpus reedi*

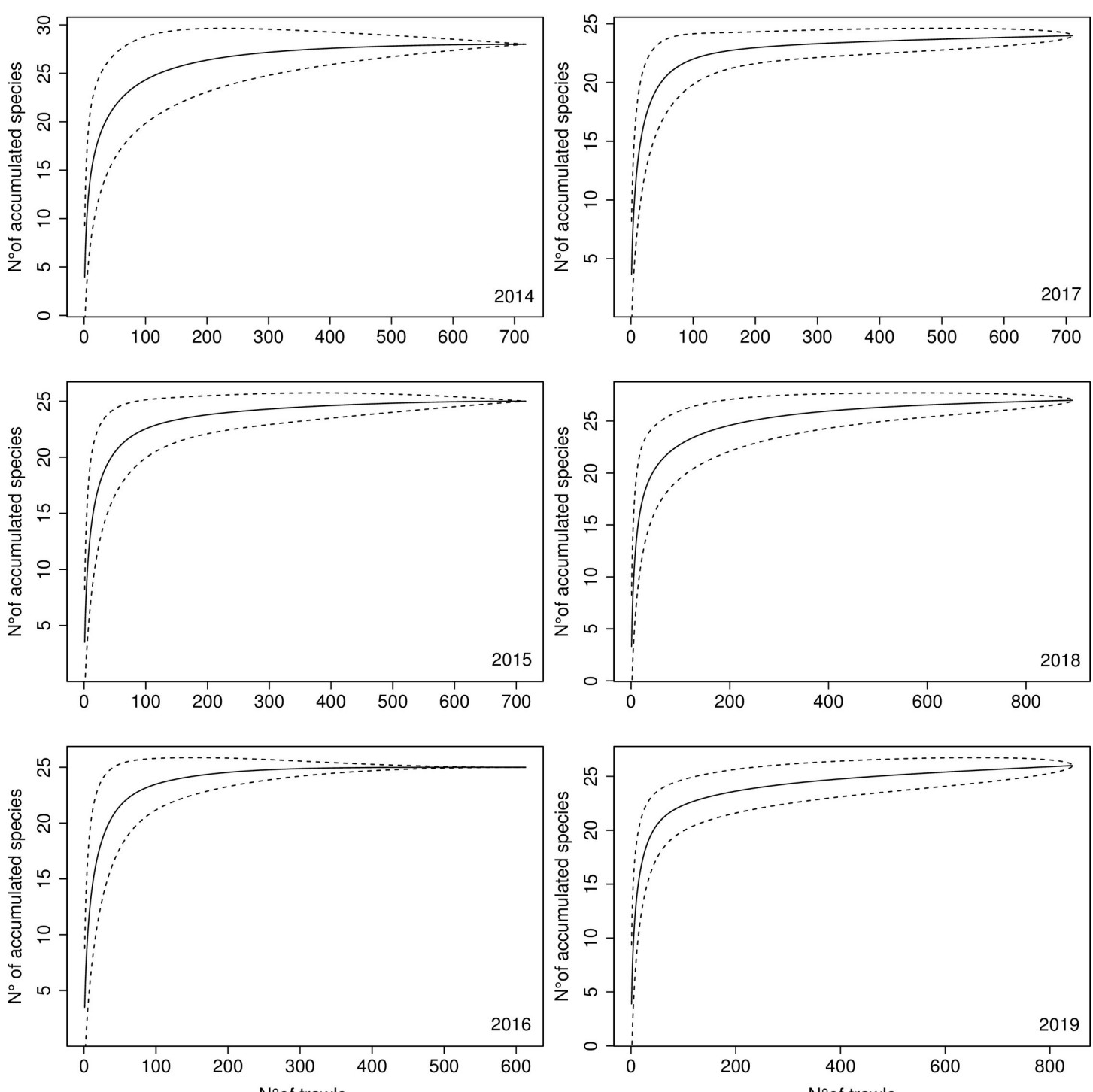

**Fig 2. Cumulative curves of species found as discards in the Chilean demersal crustacean fisheries, separated for each sampled year.** Dotted lines represent one standard deviation.

(Assemblage 2), *C. aconcagua*, *Coelorinchus chilensis*, *Psammobatis scobina*, and *Centroscyllium granulatum* (Assemblage 3), while the Assemblage 1 did not have any indicator species (Table 2).

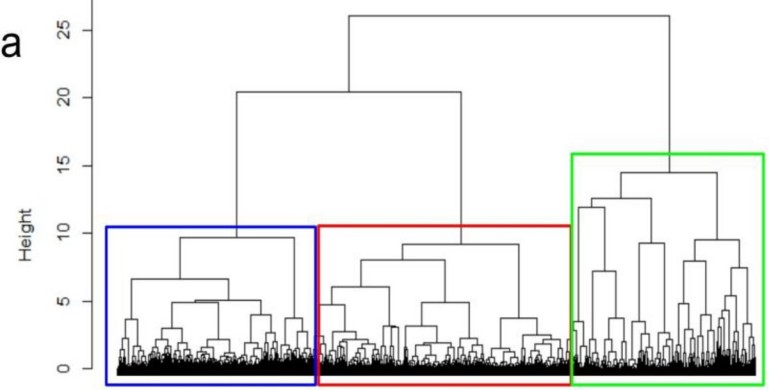

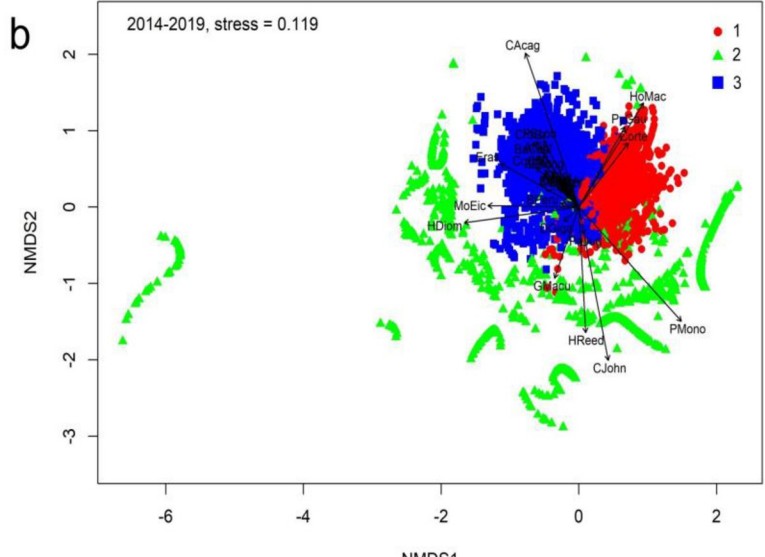

**Fig 3.** Multivariate analyses, a) beta-flexible clustering analysis and b) non-metric multidimensional scaling (nMDS) of the discarded demersal fauna from the crustacean fisheries operating off central Chile in 2014–2019. Code names are explained in Table 1.

The spatio-temporal distribution of the assemblages showed differences in depth and latitude (Fig 4). Assemblage 1 showed a shallower distribution (196 ± 56 m), with less variability (i.e., reduced standard deviation) southward 35-36˚S, where the continental shelf widens up to 65 nm (Figs 1 and 4A). Assemblage 2 showed a depth distribution around 300 m depth, with deeper and narrower distribution northward 29˚S and shallower (~180 m depth) at 36-37˚S. They were absent southward (Fig 4B). Assemblage 3 showed a deeper distribution (~400 m depth) in the northern zone, and shallower (~280 m depth) southward, with a latitudinal gradient in the depth distribution. Also, they were absent north of 29˚S (Fig 4C). All assemblages showed slight but significant differences in depth distribution at interannual scales (Table 3); assemblage 1 showed wider depth distributions in 2015 and 2018 (Fig 4D), assemblage 2 showed shallower and deeper distributions in 2018 and 2019, respectively (Fig 4E), and assemblage 3 was shallower in 2016–2017 (Fig 4F). Significant differences were observed in the

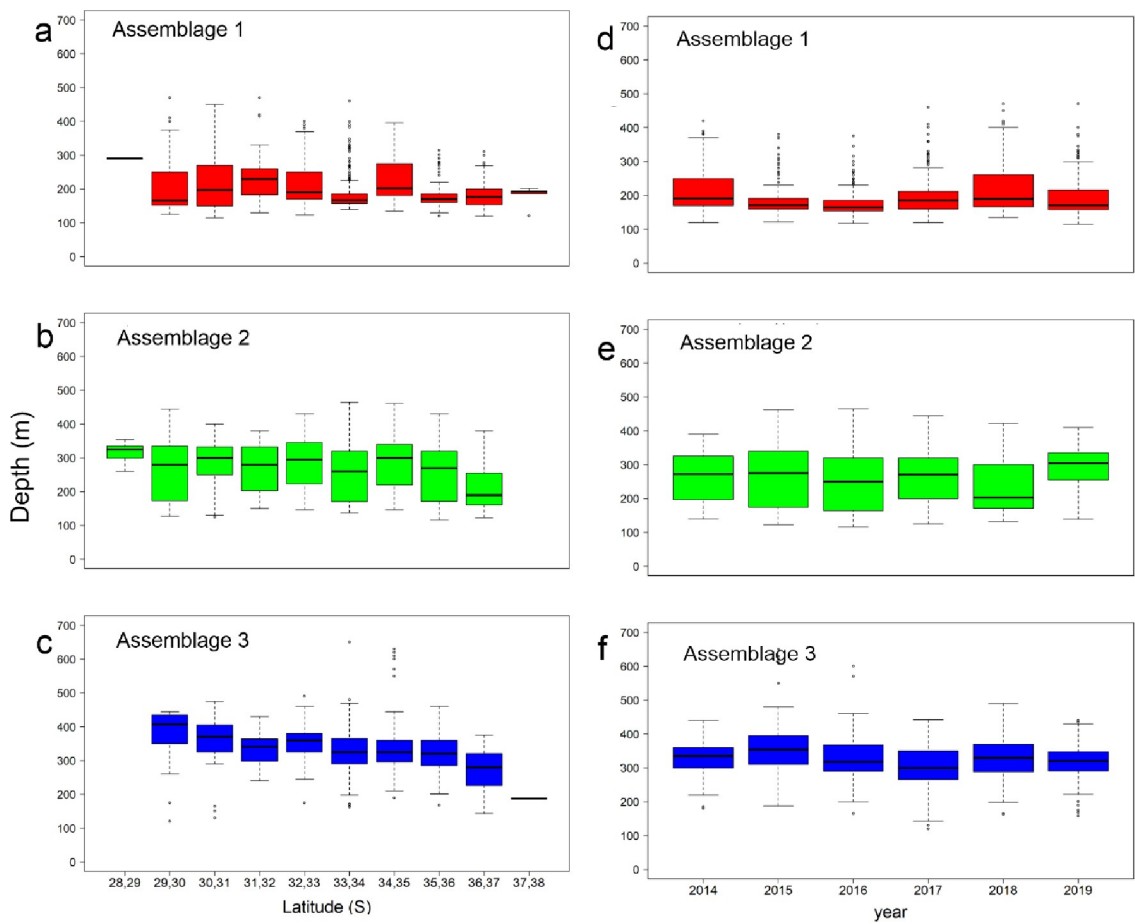

**Fig 4.** Boxplots (mean, standard deviation, maximum and minimum) of depth distribution for each assemblage identified by the multivariate analyses and defined in Table 2, by latitude (a, b, c) and year (d, e, f).

distribution depth of each assemblage, by year and latitude, although latitudinal variation was more relevant than the interannual variation (Table 3).

The PERMANOVA showed that the variables depth, latitude, year, and the interaction between year and latitude explained the differences in the composition of discarded species. The variable season and the interaction between year and depth appeared as significant ($P<0.05$) but their addition did not improve the model, as the AICc indicated (Table 4). Also,

**Table 3. Nested Analysis of Variance (ANOVA) of depth distribution of assemblages identified by multivariate analyses and defined in Table 3, compared by latitude and year.**

| | Df | SS | MS | F | P value |
|---|---|---|---|---|---|
| Depth~Assemblage+Assemblage*Latitude | | | | | |
| Assemblage | 1 | 3164942 | 3164942 | 499.21 | <2E-16 |
| Assemblage*Latitude | 1 | 210608 | 210608 | 33.22 | 8.79E-09 |
| Residuals | 4472 | 28352139 | 6340 | | |
| Depth~Assemblage+Assemblage*Year | | | | | |
| Assemblage | 1 | 3164942 | 3164942 | 497.5 | <2E-16 |
| Assemblage*Year | 1 | 113148 | 113148 | 17.79 | 0.000025 |
| Residuals | 4472 | 28449600 | 6362 | | |

**Table 4. PERMANOVA results showing the influence of all factors considered in the present study, zone, year, depth, and its interactions.** Corrected Akaike Information Criterion (AICc) also is given.

| | df | SS | $R^2$ | F | P-value |
|---|---|---|---|---|---|
| **Model 0 = Species * 1 AICc = -4725.8** | | | | | |
| Model | 0 | 0 | 0 | 0 | |
| Residual | 4474 | 1555.8 | 1 | | |
| Total | 4474 | 1555.8 | 1 | | |
| **Model 1 = Species * Depth AICc = -13579.6** | | | | | |
| Depth | 1 | 29.871 | 0.12025 | 613.85 | 0.001 |
| Residual | 4491 | 218.543 | 0.87975 | | |
| Total | 4492 | 248.415 | 1 | | |
| **Model 2 = Species * Latitude AICc = -4762.6** | | | | | |
| Latitude | 1 | 13.42 | 0.00862 | 38.913 | 0.001 |
| Residual | 4473 | 1542.4 | 0.99138 | | |
| Total | 4474 | 1555.81 | 1 | | |
| **Model 3 = Species ~Season AICc = -4744.4** | | | | | |
| Season | 1 | 7.14 | 0.00459 | 20.617 | 0.001 |
| Residual | 4473 | 1548.68 | 0.99541 | | |
| Total | 4474 | 1555.81 | 1 | | |
| **Model 4 = Species * Latitude+ Year AICc = -4797.0** | | | | | |
| Latitude | 1 | 13.42 | 0.00862 | 39.222 | 0.001 |
| Year | 1 | 12.5 | 0.00803 | 36.54 | 0.001 |
| Residual | 4472 | 1529.9 | 0.98334 | | |
| Total | 4474 | 1555.81 | 1 | | |
| **Model 5 = Species * Latitude+ Year+Latitude:Year AICc = -13090.8** | | | | | |
| Latitude | 1 | 2.253 | 0.00907 | 41.5378 | 0.001 |
| Year | 1 | 2.19 | 0.00881 | 40.3755 | 0.001 |
| Latitude x Year | 1 | 0.529 | 0.00213 | 9.7617 | 0.001 |
| Residual | 4489 | 243.443 | 0.97999 | | |
| Total | 4492 | 248.415 | 1 | | |
| **Model 6 = Species * Latitude+Year+Depth+Year:Depth AICc = -5242.2** | | | | | |
| Latitude | 1 | 13.42 | 0.00862 | 43.344 | 0.001 |
| Year | 1 | 12.5 | 0.00803 | 40.38 | 0.001 |
| Depth | 1 | 139.59 | 0.08972 | 450.923 | 0.001 |
| Year x Depth | 1 | 6.53 | 0.00419 | 21.081 | 0.001 |
| Residual | 4470 | 1383.78 | 0.88942 | | |
| Total | 4474 | 1555.81 | 1 | | |
| **Model 7 = Species * Latitude+Year+Depth+Year:Latitude AICc = -13640.5** | | | | | |
| Latitude | 1 | 2.253 | 0.00907 | 46.9542 | 0.001 |
| Year | 1 | 2.19 | 0.00881 | 45.6402 | 0.001 |
| Depth | 1 | 28.187 | 0.11347 | 587.5346 | 0.001 |
| Latitude x Year | 1 | 0.472 | 0.0019 | 9.8439 | 0.001 |
| Residual | 4488 | 215.313 | 0.86675 | | |
| Total | 4492 | 248.415 | | | |

depth explained the largest portion of the variance, while latitude, year, and their interactions had a similar and lower influence on the assemblages. The AICc indicates that the best model was the Model 7, that considered depth, latitude, year, and the interaction between year and latitude (Table 4). Under this model, depth was the factor explaining the largest portion of variance, while latitude and year had similar influence on the assemblages.

## Spatio-temporal variation of the alpha-diversity indices

The spatio-temporal patterns of the discarded demersal fauna, based on selected alpha-diversity indices, were analysed considering all discarded taxa detected in the hauls. At the interannual scale, interpolations showed noticeable variations of diversity mainly across depths (Fig 5). Richness (*S*) and biodiversity, expressed as the Shannon-Weaver and Simpson indices, showed similar patterns. In general, the highest diversity was located at deeper ranges, particularly in 2018 (below 300 m depth), although in 2019 higher diversity was found at mid-depths south of 34˚S. In some years (2014, 2016, and 2017), high richness and diversity were also found off 30˚S at ~300 m depth (Fig 5). At 37˚S, a shallow (~100 m depth) spot of high diversity was detected in 2014, moving deeper as years passed, except for 2019 when it moved to mid-depths (Fig 4). In terms of evenness, there was high interannual variability; in 2014, evenness was higher in deeper waters, except for 37-38˚S, while in 2015–2017 evenness was higher northward 33˚S. In 2018, evenness was relatively homogenous below 200 m depth, and the following year, evenness was higher in shallower waters (<200 m depth) (Fig 5).

At a monthly scale, and at the beginning of the series (2014), there was large and extended diversity during the austral summer, with lowest values of alpha diversity during the austral winter. The largest diversity indices were found northward of 30˚S and southward of 36˚S, except for 2018 and 2019 when large areas of higher diversity were detected at 31˚ and 35˚S (Fig 6). In general, those areas and times with the lowest diversity showed higher evenness (Fig 6), but also coincided with periods of positive SST anomalies (i.e., warmer waters, Fig 8). Finally, areas with higher values of evenness showed latitudinal variations at the interannual scale, moving from the north in austral summer southward in June-July (Fig 6).

## Environmental conditions

Marked seasonality in sea surface temperatures (SST) was observed along the region, with relative warmer (> 17˚C) periods during austral spring-summer and relatively colder (< 13˚C) periods during the austral autumn-winter (Fig 7). This pattern was more pronounced northward of 35˚S compared to southward of 37˚S, where SST was colder and less variable throughout the years. The SST time series showed colder autumn-winters during 2003, 2007, 2014, and 2018, while warmer spring-summers were observed in 2009, 2012, 2017, and 2020. Nevertheless, 2017 was a warmer year throughout the whole study region.

The normalised spatio-temporal EOF amplitudes are shown in Fig 8. EOF1 represented c.a. 67% of the variability, whereas EOF2 and EOF3 explained nearly 19% of SSTA. Along the region, EOF1 variability was nearly uniform though it was relatively lower between 33˚S to 37˚S, on the contrary to EOF3, which increased between these latitudes (Fig 8).

The temporal EOF1 was dominated by low-frequency (> 1 year) oscillations, especially during 2007–2009 and 2015–2018. On the other hand, the temporal EOF2 and EOF3 presented relatively high-frequency oscillations, which were dominated by intraseasonal (20 to 90 days) and monthly oscillations (30 days). The spatial pattern of SSTAr (a reconstruction using the first three EOF modes) showed marked temporal variation along the region. Between 2003–2013, semi-annual variability was observed. A colder event (below 1˚C, Fig 7) was present throughout the region between 2007 and 2008, coinciding with negative MEI (Multivariate ENSO Index version 2 (https://psl.noaa.gov/enso/mei/) values (Fig 8). A cooler period was observed between 2009 and 2015, followed by evident warmer events between 2015 and 2017. Afterwards, the SSTAr presented mainly negative anomalies (Fig 8).

The spatio-temporal series of chl-*a* concentration showed noticeable discontinuities at the latitudinal scale, with the largest values between 35-37˚S (off Itata terrace and Arauco Gulf), between 39-40˚S, and 29-30˚S (Fig 9). At a temporal scale, the highest concentrations occurred

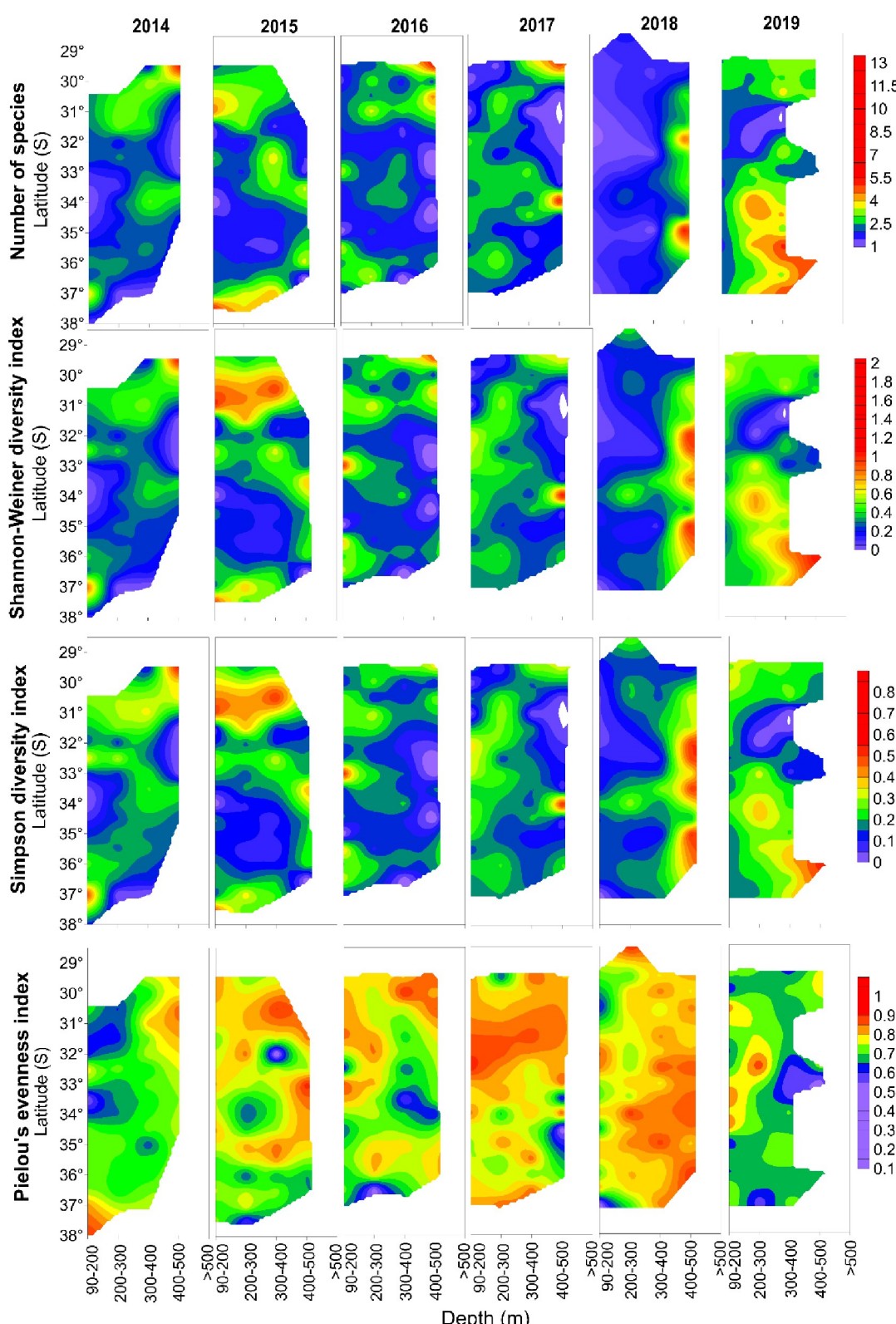

**Fig 5. Spatial variability (in latitude and depth) of alpha-biodiversity indices of the discarded demersal community off central Chile between 2014–2019.**

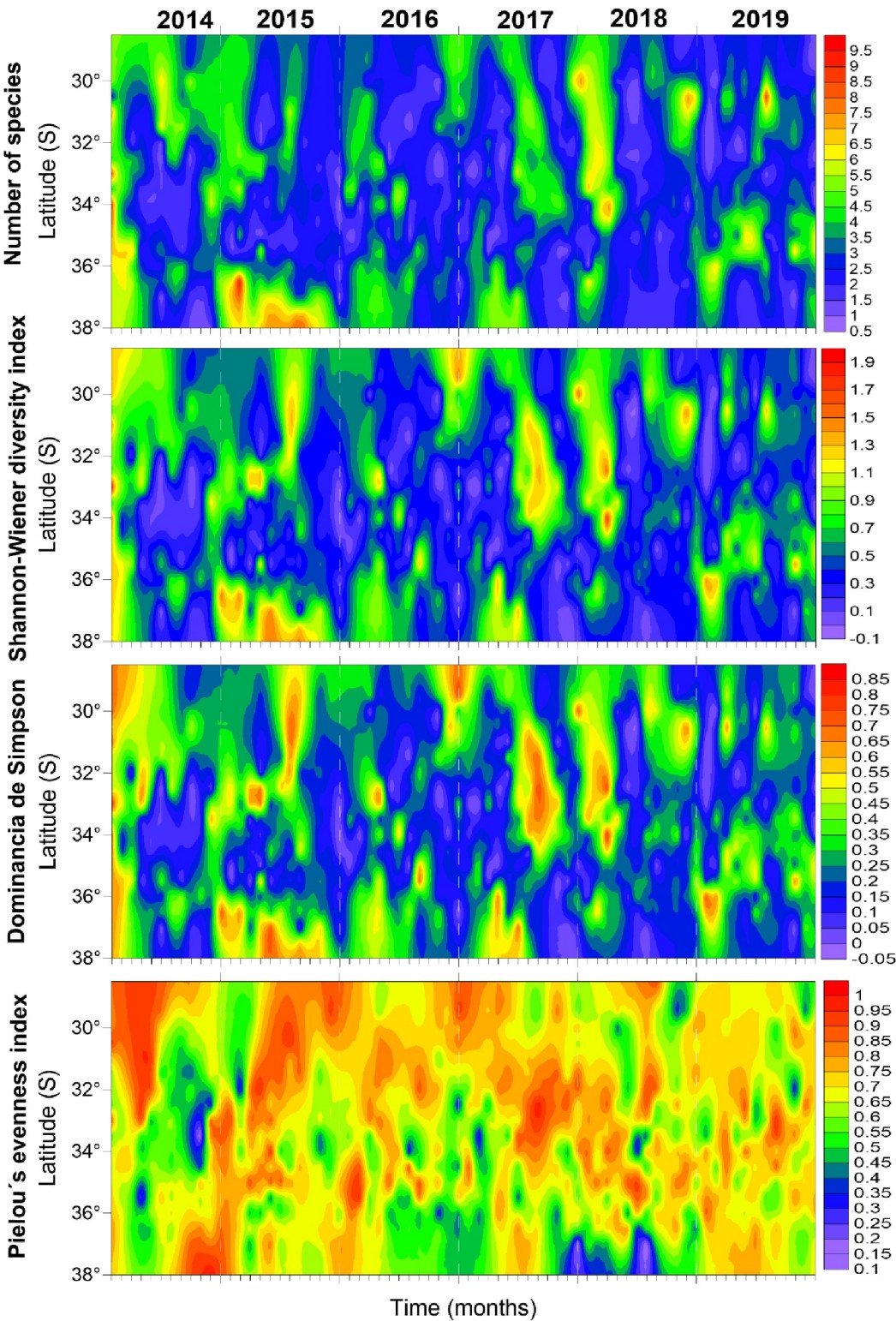

**Fig 6. Hovmöller diagram (latitude vs months between 2014–2019) of the alpha-diversity indices of the discarded faunal demersal community off central Chile.**

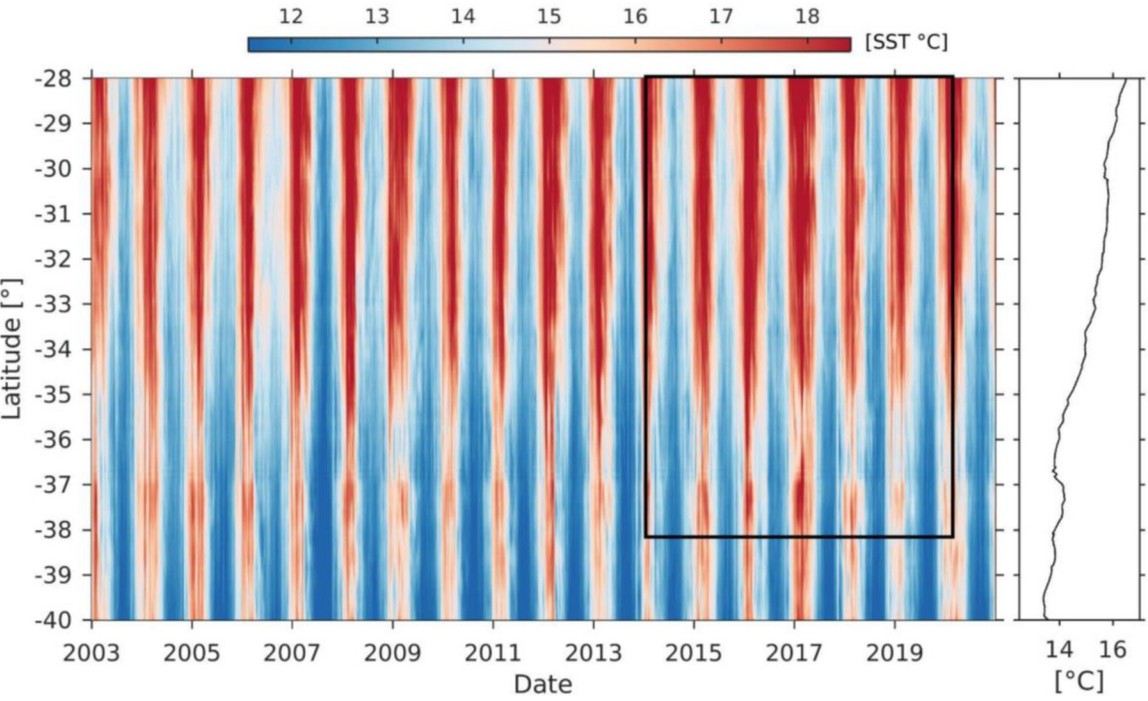

**Fig 7. Hovmöller diagram of coastal (first 100 nm) daily sea surface temperatures (SST) between 28˚S and 40˚S.** The black box indicates the spatio-temporal scale of the crustacean demersal fishery discard data. The right panel shows the mean latitudinal variation throughout the series.

during austral spring and summer. Large and extended patches of Chl-a were observed during the austral summer of 2004–2005, 2010–2011, and 2014–2015, while between 30-33˚S there was a reduced density of the standing stock (Fig 9). Contrary to the spatial average of SST (Fig 7), the Chl-*a* average increases southward in 0.125 mg m$^{-3}$ per degree of latitude.

Similar to Chl-a, meridional wind-stress (Fig 10) showed a positive slope as a function of latitude, increasing 3x10$^{-3}$ N m$^{-2}$ per degree of latitude. The wind-stress changes in the studied region from relatively high negative (4x10$^{-2}$ N m$^{-2}$) to near zero. Wind-stress north of 34˚S was mainly negative (northerly winds), while south of this latitude positive wind-stress (southerly winds) were observed, with a marked seasonality associated with austral winter.

## Modelling

The GAM models showed that the variability of the richness of the discards was explained only by Year and Depth, with highest richness is at higher depths (600 m) but with wider CI (Table 5 and Fig 11A and 11B). None of the environmental data derived from satellites showed a significant effect on alpha-diversity indices. Also, Shannon-Weiner, Simpson and Pielou's indices only showed a significant effect of Year, explaining between 5 and 10% of the variability (Fig 11C–11E and Table 5). For all the diversity indices, the interannual variations were similar, with a decrease of diversity (and increase in the evenness) during 2015 and 2016 (warm years), and an increase of diversity, but with higher variability in evenness values, in 2018 and 2019 (Fig 11). Therefore, at spatial scales of dozens of kilometres and a monthly basis, interannual variations of biodiversity occurred, but surface conditions of temperature, chlorophyll-a and wind stress did not affect the diversity of the discarded demersal fauna of the crustacean fishery operating along central Chile.

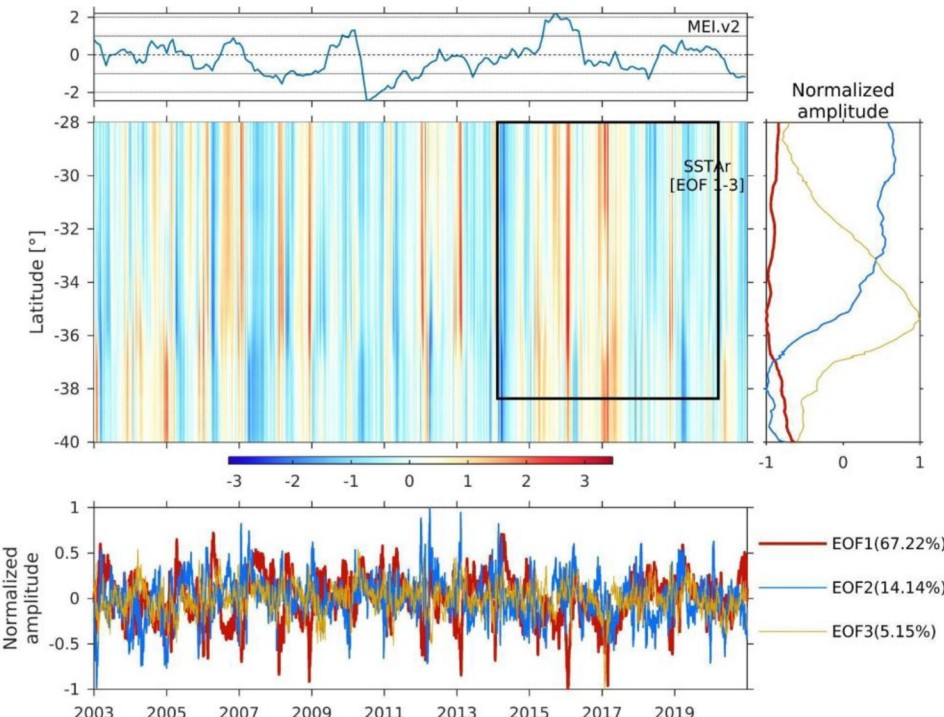

**Fig 8. Time series of multivariate ENSO index version 2 (https://psl.noaa.gov/enso/mei/) (MEI, upper panel), and Hovmöller diagram of the reconstructed sea surface temperature anomaly (SSTAr, medium panel) using the first three Empirical Orthogonal Function (EOF) modes.** The spatial EOFs (right panel), the temporal EOF (bottom panel) and their variability contribution are also shown. The black box indicates the spatio-temporal scale of the crustacean demersal fishery discards.

## Discussion

The crustacean demersal fisheries operating within the limits of the continental shelf and the upper trench, with large abundances in the depth range of 100–300 m [30], influence multi-species assemblages that vary in composition and structure according to depth. Discards from catches along the deeper portion of the continental shelf off central Chile was dominated by finfish (flounders, hakes) and crustaceans (squat lobsters) (assemblage 1); at the shelf-break, there were high abundances of hakes, squat lobsters, and deep-sea shrimps (assemblage 2). Finally, along the continental slope, hakes, grenadiers, and demersal sharks were the most frequent species (assemblage 3). The diversity of these assemblages varied interannually and was independent of the mesoscale environmental processes occurring in the mixed layer at monthly temporal scale between 2014 and 2019.

### Composition of discarded demersal faunal

The composition of discarded demersal faunal, based on the scientific observer program of the crustacean demersal fisheries, was similar to those described based on fishery-independent bottom trawl surveys [14], in terms of species found (108 *vs.* 106, respectively) and frequencies of occurrence, particularly hakes, flounders and squat lobsters. Nonetheless, some differences were detected. Montero et al. [14] described that the elasmobranchs during the period 2000–2014 showed frequencies of occurrence of 5.47 to 4.61% for *Psammobatis scobina*, *Hexanchus griseus* and *Dipturus flavirostris*. Our results showed that elasmobranchs in the period 2014–2019 occurred in higher frequency (between 13 and 10%), particularly *Aculeola nigra*, *P.*

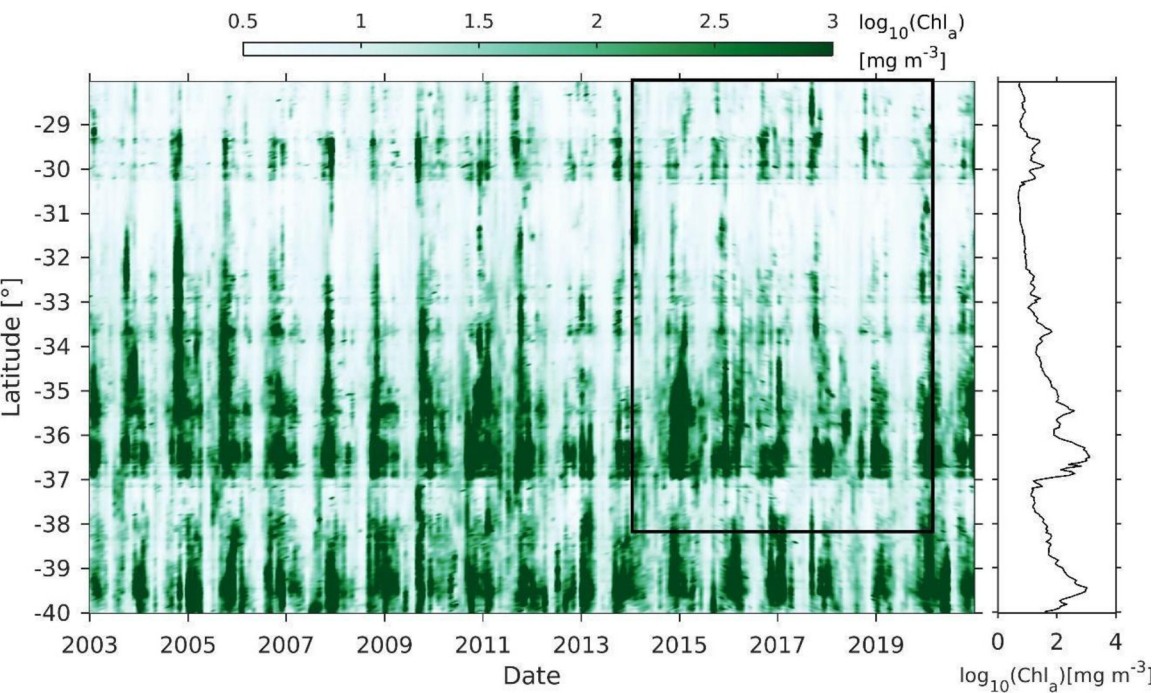

**Fig 9. Hovmöller diagram of monthly coastal (first 100 nm) log-transformed Chlorophyll-*a* (Chl-a) between 28°S and 40°S.** The black box indicates the spatio-temporal scale of the discard data of the crustacean demersal fishery. The right panel shows latitudinal variation in the mean value throughout the series.

*scobina* and *Centroscyllium granulatum*. Elasmobranch overfishing has been increasing in the last decades [40, 41]. Several other causes may be responsible for this change, such as the spatial extension (25–37°S *vs.* 28–38°S), temporal resolution (samplings mainly in austral winter *vs.* throughout the year), and fishery considered (only squat lobster *P. monodon* fishery) in the study done by Montero et al. [14].

According to the composition of the discards, the most vulnerable species was the Chilean hake, *M. gayi*, which was ubiquitous in the studied area, occurring in 95% of the hauls. Juvenile hake (less than 2 years-old) inhabit the continental shelf ($< 200$ m depth) throughout the latitudinal extent of the study [42]. After 2004, 2-year-old hake reached a proportion of 48% maturity due to a reduction in the length at maturity [43]. To reduce the chances of harvesting juveniles during the recruitment process, it is highly recommended to restrict crustacean and hake fleet bottom-trawling operations to areas deeper than 200 m depth [42].

## Spatial distribution of demersal assemblages of discards

The depth-diversity gradient has been previously described for the demersal fauna off Chile, with the highest richness detected at intermediate depths (300–400 m) and the lowest in shallow waters (50–100 m), based on annual squat lobster data [18]. This strongly suggests that observed changes in the depth distribution of discarded fauna diversity between 2014 and 2019 are a proxy of the demersal fauna community. The vertical spatial pattern of diversity shows similarities to other areas around the globe, caused by specific topographic conditions [44], zonation [45, 46] or the continuous replacement of species [47].

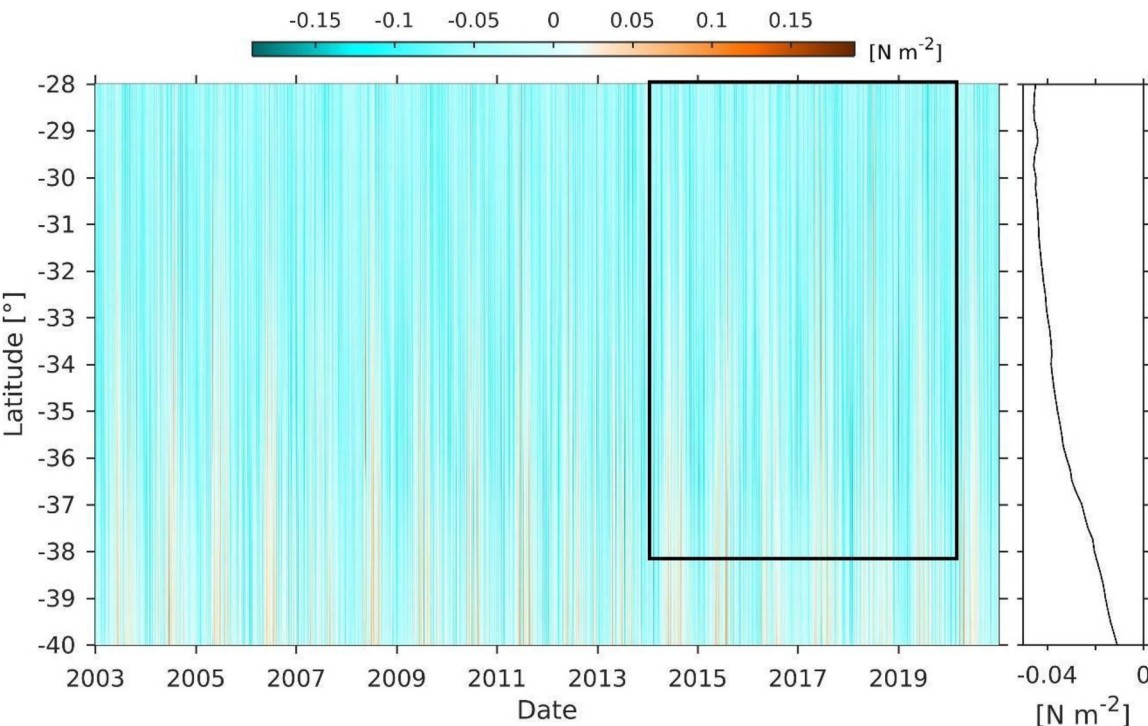

**Fig 10. Hovmöller diagram of daily-averaged coastal (100 nm) northerly (red)-southerly (blue, upwelling-favourable) wind-stress (N m$^{-2}$) between 28°S and 40°S.** The black box indicates the spatio-temporal scale of the discard data of the crustacean demersal fishery. The right panel shows latitudinal variation in the mean value throughout the series.

## Interannual variability of assemblages and diversity of discards

Observer data from a sample of fishing effort from fisheries with partial coverage rates are routinely raised to produce fleet-wide estimates of bycatch or discard. Therefore, bycatch observer data can be assessed to determine if temporal trends in catch levels and nominal catch rates are occurring [12] and the temporal variation in the composition of the discards (this study).

**Table 5. Main results of the GAM modelling for alpha-diversity indices of discarded faunal demersal community.**

| Co-variable | Df | Resid. Dev | Deviance Resid. | F | Pr(>F) | AIC | Pseudo R$^2$ |
|---|---|---|---|---|---|---|---|
| GAM (spline) | Richness | | | | | | |
| Null | 282 | 353.74 | | | | 1150.775 | |
| Year | 277 | 312.88 | 40.86 | 6.22 | 3.82E-06 | 1119.917 | 11.6 |
| Depth (m) | 272 | 295.97 | 16.91 | 5.91 | 0.01569 | 1111.006 | 4.8 |
| GAM (spline) | Shannon-Weiner diversity index | | | | | | |
| Null | 282 | 50.62 | | | | 320.0478 | |
| Year | 277 | 45.12 | 5.49 | 3.71 | 5.91E-06 | 297.5175 | 0.1 |
| GAM (spline) | Simpson diversity index | | | | | | |
| Null | 282 | 12.18 | | | | -83.0904 | |
| Year | 277 | 11 | 1.17 | 5.91 | 3.25E-05 | -101.8019 | 0.1 |
| GAM (spline) | Pielou's evenness index | | | | | | |
| Null | 222 | 7.57 | | | | -117.3084 | |
| Year | 277 | 7.2 | 0.37 | 2.26 | 4.90E-02 | -101.8019 | 0.05 |

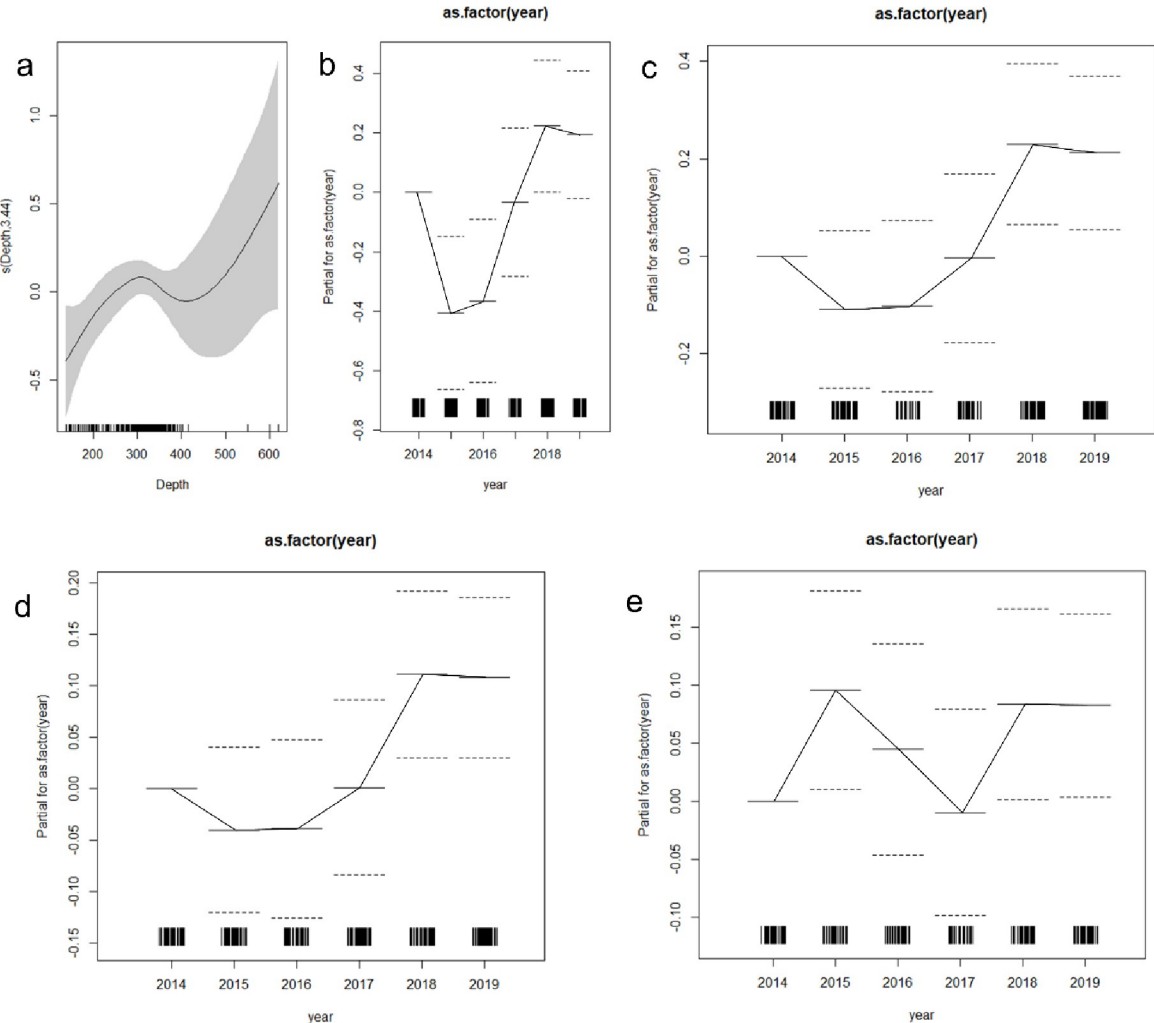

**Fig 11.** Effects of the predictors Depth and Year on richness (a and b, respectively), fitted for a GAM model. Effects of predictor Year on the Shannon-Weiner diversity index (c), Simpson index (d) and Pielou's evenness (e). Straight lines represent the mean and dotted lines represent the confidence interval (CI).

Although demersal species change in abundance and distribution are expected to occur more slowly than pelagic species, due to slower growth rates, larger life cycle and body sizes [48, 49], the interannual variation in abundance and composition of discards may be caused by large, basin-scale environmental processes. Indeed, we observed significant among-years differences in the depth distribution of each assemblage, suggesting behaviour and movement responses are likely key to define depth distributional limits. Although environmental conditions were included in our analyses, no significant effects were identified, probably caused by the short time series (only six years), and the scale of processes studied (local rather than regional or basin-scale), or perhaps a combination of the a) mismatch between the scale/timing of fishing and that of time- and space-averaged environmental conditions (i.e., they are not necessarily simultaneous), and b) the potential that surface conditions do not by themselves consistently match key variability of environmental change occurring at depth. In terms of biodiversity, events such as "Godzilla" ENSO and the coastal ENSO, occurring during austral summer 2015–2016 and 2016–2017, respectively, correlated with reduced diversity of the

demersal discards. During the "normal" years 2018 and 2019, spatial patterns of biodiversity changed, with low diversity occurring at the shelf, and even dominance was relatively low, suggesting little amounts of species. Nonetheless, all the diversity and evenness were deep off the slope.

## Biophysical interactions

The environmental changes along the region are consistent with a marked shift in the coastal conditions. The latitudinal averages of SST showed a negative slope of -0.17˚C per degree of latitude (Fig 6). On the other hand, a positive slope was observed in meridional wind-stress (northerly), $3 \times 10^{-3}$ N m$^{-2}$ per degrees of latitude (Fig 4), similar to Chl-a, with a slope of 0.125 mg m$^{-3}$ per degrees of latitude (Fig 8). The southward latitudinal pattern on wind-stress was consistent with similar obtained by Figueroa and Moffat [25] using another source of satellite wind data. Also, the thermal conditions in the region report an inner-shelf coastal shift [50]. The study region presented high kinetic energy southward of 30˚S, suggesting the presence of a coastal transition zone off the coast [51]. In addition, the region presented a high mesoscale activity which has been linked with the extension to oceanic conditions of copepods assemblages forced by physical/biological mechanisms [52, 53]. In productivity, a spatial shift in this region has also been described from 20 years of satellite data [54].

The spatial patterns of the assemblages, particularly assemblage 2 and 3, showed similarities with a latitudinal change in SST and wind-stress, and latitudinal increases in Chl-a concentrations. Ekman transport and Ekman pumping may act synchronously to extend and produce highest Chl-a values during spring-summer [53]. This high productivity is the most significant carbon contribution towards deep waters, acting as a pelagic-benthic coupling in central Chile [19]. Additionally, the topographic effect caused by the meridional changes of the coastline orientation, dominates the wind-induced upwelling in several parts of the Chilean coast, namely at 30˚S, 33˚S and 37˚S [25], observed in the wind-stress and Chl-*a* concentration patterns during the studied period. In other eastern boundary currents, such as the Benguela current [55], latitudinal, or along-shore variations in the demersal fauna from eastern boundary currents may be related to upwelling events, like in the Benguela current [55].

At larger spatial and temporal scales, sea surface temperatures, ENSO, and PDO may have impacts at population level of many species; for example, these variables negatively influence population growth of the cardinalfish *Epigonus crassicaudus* living at depths of 300–550 m [29]. During ENSO years, an increase of the geographic distribution of the hake occurs, lowering their total density and decreasing the mortality by cannibalism [56, 57].

Although spatio-temporal changes in the assemblages and the diversity of the discards were found, we were unable to detect biophysical interactions with the mesoscales processes occurring in the south HCS. This may be due to the spatial scales considered to analyse satellite data: grids of 27 x 27 km were used to analyse the environmental data (SST, SSTA, Chl-a, wind-stress), while biological data (e.g., composition and CPUA of discards) change longitudinally at faster rates, particularly at the shelf-break and slope, where depth is the main factor structuring the assemblages. Additionally, at temporal scales, the monthly resolution of the sampling reduced the influence of physical processes occurring at shorter time scales, such as upwelling events, inertial and sub-inertial processes [58], and its biological consequences, such as primary productivity.

Therefore, it is mandatory to obtain more accurate *in situ* oceanographic conditions during the commercial hauls (e.g., temperature sensors on the trawler), in order to determine the real influence of those conditions in the spatio-temporal variations of the demersal community living along the Humboldt Current System.

Finally, the presence of sessile organisms, such as *Hormatia*, in the discards suggest that the fishery activity may have impacting on temperate mesophotic reefs, particularly those occurring in the deepest mesophotic zone (90–110 m depth) along central Chile. Most of the trawling activity of the crustacean demersal fisheries occurs on sandy and muddy bottoms [30], but the presence of anthozoans in the discards may be an opportunity for developing conservation measurements in the fishery, to preserve these reefs.

## Supporting information

**S1 File. List of all taxonomic groups identified as discards of the Chilean crustacean demersal fisheries, operating between 2014 and 2019.**
(DOCX)

## Acknowledgments

We appreciate the comments and suggestions by Dr. P. Saenz-Agudelo (Universidad Austral de Chile), and by Dr. Jorge Páramo (Universidad de Magdalena, Colombia) and Dr. Taner Yildiz to an early version of the manuscript.

## Author Contributions

**Conceptualization:** Mauricio F. Landaeta, Carola Hernández-Santoro, Sergio A. Navarrete, Alejandro Pérez-Matus.

**Data curation:** Claudio Bernal.

**Formal analysis:** Mauricio F. Landaeta, Carola Hernández-Santoro, Francesca V. Search, Manuel I. Castillo.

**Funding acquisition:** Mauricio F. Landaeta, Sergio A. Navarrete, Evie A. Wieters, Ricardo Beldade, Alejandro Pérez-Matus.

**Supervision:** Claudio Bernal.

**Validation:** Manuel I. Castillo.

**Visualization:** Manuel I. Castillo.

**Writing – original draft:** Mauricio F. Landaeta.

**Writing – review & editing:** Mauricio F. Landaeta, Claudio Bernal, Sergio A. Navarrete, Evie A. Wieters, Ricardo Beldade, Ana Navarro Campoi, Alejandro Pérez-Matus.

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
