## [Decision Letter · Decision Letter 0]

26 Dec 2022

PONE-D-22-26514Spatio-temporal patterns of the crustacean demersal fishery discard from the south Humboldt Current System, based on scientific observer program (2014-2019)PLOS ONE

Dear Dr. Landaela,

Thank you for submitting your manuscript to PLOS ONE. After careful consideration, we feel that it has merit but does not fully meet PLOS ONE’s publication criteria as it currently stands. Therefore, we invite you to submit a revised version of the manuscript that addresses the points raised during the review process.

We want to apologise the authors for the delay in response to the submission, but it was incredibly hard to allocate good reviewers for the manuscript. However, I am confident that the results are worth the effort. Please submit your revised manuscript by Feb 09 2023 11:59PM. If you will need more time than this to complete your revisions, please reply to this message or contact the journal office at plosone@plos.org. Please include the following items when submitting your revised manuscript:A rebuttal letter that responds to each point raised by the academic editor and reviewer(s). You should upload this letter as a separate file labeled 'Response to Reviewers'.A marked-up copy of your manuscript that highlights changes made to the original version. You should upload this as a separate file labeled 'Revised Manuscript with Track Changes'.An unmarked version of your revised paper without tracked changes. You should upload this as a separate file labeled 'Manuscript'.If applicable, we recommend that you deposit your laboratory protocols in protocols.io to enhance the reproducibility of your results. Protocols.io assigns your protocol its own identifier (DOI) so that it can be cited independently in the future. For instructions see: https://journals.plos.org/plosone/s/submission-guidelines#loc-laboratory-protocols. Additionally, PLOS ONE offers an option for publishing peer-reviewed Lab Protocol articles, which describe protocols hosted on protocols.io. Read more information on sharing protocols at https://plos.org/protocols?utm_medium=editorial-email&utm_source=authorletters&utm_campaign=protocols.

We look forward to receiving your revised manuscript.

Kind regards,

José M. Riascos, Ph.D.

Section Editor

PLOS ONE

Journal Requirements:

2. In your Methods section, please provide additional information regarding the permits you obtained for the work. Please ensure you have included the full name of the authority that approved the field site access and, if no permits were required, a brief statement explaining why

"This work was partially funded by the Millenium Nucleus for Ecology and Conservation of Temperate Mesophotic Reef Ecosystems (NUTME) grant NCN19_056 to APM. We appreciate the comments and suggestions by Dr. P. Saenz-Agudelo (Universidad Austral de Chile)."

"This work was partially funded by the Millenium Nucleus for Ecology and Conservation of Temperate Mesophotic Reef Ecosystems (NUTME) grant NCN19_056 to APM, EAW, SAN, RB, PSA and MFL"

Reviewers' comments:

Reviewer's Responses to Questions

**Comments to the Author**

1. Is the manuscript technically sound, and do the data support the conclusions?

Reviewer #1: Yes

Reviewer #2: Yes

2. Has the statistical analysis been performed appropriately and rigorously? 

Reviewer #1: Yes

Reviewer #2: Yes

3. Have the authors made all data underlying the findings in their manuscript fully available?

Reviewer #1: Yes

Reviewer #2: Yes

4. Is the manuscript presented in an intelligible fashion and written in standard English?

Reviewer #1: Yes

Reviewer #2: Yes

5. Review Comments to the Author

Reviewer #1: The manuscript is very interesting because it analyzes the spatio-temporal patterns of the crustacean demersal fishery discard from the south Humboldt Current System, based on scientific observer program. Some corrections and analysis are required before being accepted to publish.

Line 69: change ... demersal crustacean fisheries by ... demersal crustacean chilean fisheries.

Delete table 1 and replace with a species accumulation curves using the samples-based rarefaction method for each year.

Line 127: Differences between industrial and artisanal fleets are also related to engine power and fishing gear dimensions. Please evaluate if there are differences between the industrial and artisanal fleets with respect to engine power and the dimensions of the fishing gear.

Figure 3: correct the names in the X and Y axis that are cut off.

Reviewer #2: The article evaluates detailed data on data collection by observers in the problem of discard, which is an important problem in fisheries. The applied statistics were chosen correctly. however, some additions are required for a more understandable flow. The parts that need to be added are indicated on the text.

6. PLOS authors have the option to publish the peer review history of their article (what does this mean?). If published, this will include your full peer review and any attached files.

Reviewer #1: **Yes: **Jorge Paramo, Universidad del Magdalena, Colombia

Reviewer #2: **Yes: **Taner Yildiz

---

## [Author Response · Author response to Decision Letter 0]

26 Jan 2023

Reviewer #1: The manuscript is very interesting because it analyzes the spatio-temporal patterns of the crustacean demersal fishery discard from the south Humboldt Current System, based on scientific observer program. Some corrections and analysis are required before being accepted to publish.

Line 69: change ... demersal crustacean fisheries by ... demersal crustacean chilean fisheries.

Resp. The change was done accordingly.

Delete table 1 and replace with a species accumulation curves using the samples-based rarefaction method for each year.

R. Table 1 was deleted following the suggestions by the reviewer. Also, we carried out the species accumulation curves by year.

Line 127: Differences between industrial and artisanal fleets are also related to engine power and fishing gear dimensions. Please evaluate if there are differences between the industrial and artisanal fleets with respect to engine power and the dimensions of the fishing gear.

R. In this version, we included the mean engine power of each fleet, including a statement indicating that the fishing gear was the same in dimensions for artisanal and industrial fleets.

Figure 3: correct the names in the X and Y axis that are cut off.

R. Figure 3, now Figure 4, was corrected and improved.

Reviewer #2: The article evaluates detailed data on data collection by observers in the problem of discard, which is an important problem in fisheries. The applied statistics were chosen correctly. however, some additions are required for a more understandable flow. The parts that need to be added are indicated on the text.

Line 126: commercial bottom trawler??? please clarify!

R. the explanation was included in the new version.

Line 123: please provide more informtion about the temporal scale (open period, closed season etc) of the fishery(ies)!

R. We included a new sentence “The Chilean crustacean fisheries are open throughout the year, except for September, when a reproductive ban occurs, since 2015.”

Line 127: I wonder if a metier analysis has been made for the region, can you give some information?

R. Until now, only a metier analysis has been made recently for the cardinalfish (Epigonus crassicaudus) fishery (Cubillos et al. 2022). However, those type of analysis still is lacking for the Chilean demersal crustacean fisheries. 

Line 127: specify the fishing gear?

R. Now, we have included specifications of the fishing gear.

Line 127: use same mesh size in the cod-end? if they use bottom trawl?

R. Yes, all the fleet uses the same gear and mesh size, according to Chilean laws. This information is given in the new version of the ms.

Line 129: I wonder if there is a trend related to the daily vertical distribution of Merluccius gayi regarding night hauls, because it will definitely show itself in the discard.

R. Hauls were made during night (21:00 to 02:00 h) and day hours (09:00 to 20:00 hrs), but irrespective of the time of the day, the occurrence of M. gayi in the hauls was always high (95%).

Line 129: Can you provide information about haul duration such as minimum and maximum values?

R. This information (minimum, maximum, mean, one SD) was included in the new version.

Line 138: Who decided to discard frction of catch, fishers? please specified!

R. According to the Chilean laws, only the catches of the target species can be landed, and therefore, non-target species are discarded, e.g., thrown away at sea.

Line 138: These species are always discarded?

R. Yes, always.

Line 139: please provide more detailed information about this program and detail the observation processor!

R. We included a paragraph with a link to a web page, where there is more detailed information about this program, leaded by Instituto de Fomento Pesquero (IFOP, Chile).

---

## [Editor Report · Decision Letter 1]

6 Feb 2023

Spatio-temporal patterns of the crustacean demersal fishery discard from the south Humboldt Current System, based on scientific observer program (2014-2019) PLOS ONE

PONE-D-22-26514R1

Dear Dr. Landaela,

We’re pleased to inform you that your manuscript has been judged scientifically suitable for publication and will be formally accepted for publication once it meets all outstanding technical requirements.

Kind regards,

José M. Riascos, Ph.D.

Section Editor

PLOS ONE
---

## [Editor Report · Acceptance letter]

8 Feb 2023

PONE-D-22-26514R1 

Spatio-temporal patterns of the crustacean demersal fishery discard from the south Humboldt Current System, based on scientific observer program (2014-2019) 

Dear Dr. Landaeta:

I'm pleased to inform you that your manuscript has been deemed suitable for publication in PLOS ONE. Congratulations! Your manuscript is now with our production department. 

Kind regards, 

on behalf of

Professor José M. Riascos 

Section Editor

PLOS ONE